# Diversity and Ecosystem Services of Trichoptera

**DOI:** 10.3390/insects10050125

**Published:** 2019-05-01

**Authors:** John C. Morse, Paul B. Frandsen, Wolfram Graf, Jessica A. Thomas

**Affiliations:** 1Department of Plant & Environmental Sciences, Clemson University, E-143 Poole Agricultural Center, Clemson, SC 29634-0310, USA; 2Department of Plant & Wildlife Sciences, Brigham Young University, 701 E University Parkway Drive, Provo, UT 84602, USA; paul_frandsen@byu.edu; 3Data Science Lab, Smithsonian Institution, 600 Maryland Ave SW, Washington, DC 20024, USA; 4BOKU, Institute of Hydrobiology and Aquatic Ecology Management, University of Natural Resources and Life Sciences, Gregor Mendelstr. 33, A-1180 Vienna, Austria; wolfram.graf@boku.ac.at; 5Department of Biology, University of York, Wentworth Way, York Y010 5DD, UK; jessicaathomas@gmail.com

**Keywords:** caddisfly, trait, evolution, phylogeny, angiosperm

## Abstract

The holometabolous insect order Trichoptera (caddisflies) includes more known species than all of the other primarily aquatic orders of insects combined. They are distributed unevenly; with the greatest number and density occurring in the Oriental Biogeographic Region and the smallest in the East Palearctic. Ecosystem services provided by Trichoptera are also very diverse and include their essential roles in food webs, in biological monitoring of water quality, as food for fish and other predators (many of which are of human concern), and as engineers that stabilize gravel bed sediment. They are especially important in capturing and using a wide variety of nutrients in many forms, transforming them for use by other organisms in freshwaters and surrounding riparian areas. The general pattern of evolution for trichopteran families is becoming clearer as more genes from more taxa are sequenced and as morphological characters are becoming understood in greater detail. This increasingly credible phylogeny provides a foundation for interpreting and hypothesizing the functional traits of this diverse order of freshwater organisms and for understanding the richness of the ecological services corresponding with those traits. Our research also is gaining insight into the timing of evolutionary diversification in the order. Correlations for the use of angiosperm plant material as food and case construction material by the earliest ancestors of infraorder Plenitentoria—by at least 175 Ma—may provide insight into the timing of the origin of angiosperms.

## 1. Introduction

The caddisflies, or Trichoptera, are one of the holometabolous orders of insects for which eggs, larvae, and pupae are especially abundant and diverse in freshwater while adults are generally aerial and terrestrial. Recent reviews of caddisfly structure and biology were provided by [1,2]. Briefly, most species have one generation each year (being univoltine) and are egg-laying (oviparous). Females generally lay 30 to 1000 eggs that are deposited in a layer encased in a cement-like matrix or in a mass covered by a sticky polysaccharide called “spumaline,” which helps protect the eggs from predation. The shape of the egg mass is a characteristic of different taxa. These egg masses may be laid under water by females that crawl or dive and swim to the bottom or are deposited above the water so that emerging larvae may be washed into the waterway by rain or rising surface water. Eggs hatch in a few days or weeks and the insects spend most of their lives as larvae. Larvae molt as they grow, usually completing five growth stages (instars) before pupation, although larvae of a few taxa may have additional instars. Larvae of different taxa are free living and mobile or construct various kinds of stationary retreats or portable cases and they feed in a wide variety of ways. When mature, larvae prepare to pupate by constructing stationary shelters or modifying their portable cases into stationary shelters, often spinning various types of cocoons inside them, and then molting to the pupal form. As they complete this molt, the sclerites of the final larval instar are shed and usually stored at the posterior end of the cocoon. Pupation is completed after a few weeks, with adult emergence occurring often at specific times of day or night and at predictable times of year, cued by daylight changes, temperature, or other environmental conditions. The mature pupa, with its adult form nearly complete inside the pupal skin (exuviae), cuts through the silken cocoon and anterior end of the shelter with its prominent mandibles to exit them. It swims to the water surface or crawls onto the shore or other emergent substrate and quickly sheds its pupal skin, spreads its wings, and flies up into the protective riparian rocks and vegetation. Caddisfly adults resemble moths, but have hair on the wings, rather than scales (Figure 1), or hair with only a few scales. They are generally obscure, usually active after dark, so that they are rarely seen or recognized by laypersons, being observed mostly by fly fishers or by people near water who see them when adults emerge in large numbers. Adults have rudimentary mandibles incapable of biting and ingesting solid food, but instead each has a highly modified, spongy lower lip (labium) capable of drawing sugary liquids (e.g., nectar and honeydew) into the digestive system, providing energy to sustain adult life longer than in some other insects such as mayflies and stoneflies. Adult males and females may find each other by smell, sight, or vibrations in solid substrates. Males and females of many species produce chemical pheromones that are detected by sensory organs on the antennae, allowing potential mates to track each other through the air. Males of some species have distinctive flight patterns that entice females to meet them in the air and then to fly together to riparian habitats. In some families, a male and a female have a “hammer” on the underside of the end of the abdomen that can be used to tap the substrate in distinctive ways, alerting them that a mate is nearby. Mating usually takes place while standing on a solid surface with the male and female facing in opposite directions. The eggs are then fertilized and the life cycle is repeated.

The purpose of the present contribution is to explore the evolutionary diversification and phylogenetic relationships of caddisflies according to most recent evidence and to review and interpret the phylogeny for evolutionary trends of functional traits and the corresponding ecological services they provide.

## 2. Diversity

With 16,266 extant species worldwide (Table 1), Trichoptera are the seventh most speciose order of all insects [3] with more species than the combined total of all the other primarily aquatic insect orders: mayflies (Ephemeroptera, 3436 spp.), dragonflies and damselflies (Odonata, 5956 spp.), stoneflies (Plecoptera, 3562 spp.), and dobsonflies and alderflies (Megaloptera, 350 spp.) [4]. Only the primarily terrestrial orders Coleoptera (16,604 spp.) and Diptera (51,197 spp.) have more known freshwater species [4,5]. The extant caddisfly species are classified in 618 genera of 51 families of two suborders: Annulipalpia and Integripalpia (Table 1). In addition, there are 765 fossil species, some of which are included in 121 fossil genera and 10 fossil families (Table 1). Among these fossils, 265 species are fossilized caddisfly cases (ichnospecies) that have been assigned to 11 ichnogenera but generally not to any particular family [5].

The Trichoptera World Checklist divides the world into eight biogeographic regions: Antarctic, Afrotropical (sub-Saharan Africa), Australasian (Australia, New Zealand, New Guinea, and smaller southwest Pacific islands), East Palearctic (northern Asia), Nearctic (North America and northern Mexico), Neotropical (southern Mexico, Central America, and South America), Oriental (southern Asia), and West Palearctic (Europe and Mediterranean region). No Trichoptera have been reported from Antarctica, but among the seven other global biogeographic regions, the number of families is similar, ranging from 24 in the Afrotropical and Neotropical Regions to 32 in the Oriental Region (Table 1). However, the extant species of caddisflies are distributed unevenly among these regions. By far, the Oriental Region (southern Asia) is home to the greatest number (5854 spp.) and the greatest density (405 spp./Gm^2^); the neighboring East Palearctic Region has the fewest (1200 spp.) and the lowest density (43 spp./Gm^2^) (Table 1) [5,13].

A recent review of reports concerning insect declines, especially in well-studied parts of the world (e.g., Europe and North America) indicated that caddisfly species are being lost at a greater rate than for other freshwater insect orders. Approximately 74% of the Trichoptera species cited in these reports are in decline or are extinct [14]. The principal causes for this loss have been habitat loss (including conversion to urban and agricultural uses), pollution (mainly by application of fertilizers and pesticides), pathogens and introduced species, and climate change [14].

## 3. Ecosystem Services

Cummins [15] began cataloging habitat, habit, and trophic relationships (or feeding strategies) among aquatic insects in an effort to provide enhanced insight concerning freshwater community structure and function, to refine ecological characterization and modeling programs, and to begin identifying probable causes of degraded freshwater ecosystems. Similar syntheses of ecological diversity for larvae of Trichoptera were assembled shortly thereafter [16,17]. General characteristics in these three trait categories were then listed for North American families and genera of aquatic insects in successive editions of the widely used text reference by Merritt and Cummins [18,19,20] and Merritt et al. [21]. Holzenthal et al. [1] summarized these traits for world Trichoptera families. Recently, functional traits have been catalogued for European Trichoptera in great detail for a large number of biological and environmental parameters and at the species level [12], emphasizing the diversity of functional traits known and suspected for caddisflies. In these references, the habitats, habits, and trophic relationships of caddisflies have been shown to be especially diverse, rivaling the functional diversity of Diptera [21]. Actually, when considering the greater species diversity of aquatic/semiaquatic Diptera, the relative diversity of functional traits among known Trichoptera species is much greater. For example, all of the habitats and feeding categories catalogued for aquatic insects by these authors and nearly all of the habits are employed by at least some species of Trichoptera. Our increasing knowledge of the diversity of functional traits contributes to understanding the many different ways that caddisflies provide ecological services in freshwater ecosystems.

Ecological services provided by Trichoptera include their essential roles in food webs, in biological monitoring of water quality, and as food for fish and other predators of human concern [4]. They also have been credited with engineering their habitats and resolving forensic questions and their silk is serving as a model for exciting potential applications in material science, textiles, and medicine. In food webs, as for other groups of aquatic insects, various caddisfly larvae consume organic matter that is too large for some community members and too small for others. For example, as dead plant matter (and occasionally dead animals) from riparian and aquatic sources become entrained in the water, shredding detritivores crush and pare larger decaying organic matter (greater than 10μ^3^, “coarse particulate organic matter” or CPOM) with their mandibles into ingestible-size particles, digest the colonizing microbiota on those particles, and assimilate the resulting nutrients [22,23,24]. Larvae of a few caddisflies are shredding or piercing herbivores that crush or suck and digest living plant tissue [25,26,27,28] and also release feces in the water. Particles lost in the shredding process or released as feces become available for detritivores capable of collecting the tiny particles (less than 10μ^3^; “fine particulate organic matter” or FPOM), gathering it from the substrate or filtering it from suspension in the water [29]. Caddisfly filtering activity removes FPOM from suspension, regulating its transport downstream [30,31]. For example, a filter net of *Hydropsyche siltalai* Döhler, 1963, has an area of 38 mm^2^ and is capable of filtering 492 litres of water daily [32]. In eutrophic conditions, larvae of tolerant species such as *Hydropsyche contubernalis* McLachlan, 1865, can occur in great densities, up to 10,000 specimens/m^2^ [33], with potential for capturing and removing substantial amounts of organic matter from suspension. The bodies of all of these primary consumers become food for predators that also release feces in the environment. In this way, caddisfly larvae and other insects, with their specialized feeding strategies, access available nutrients in all their forms and in turn help ensure that nutrients are available for a wide range of inhabitants with different trophic niches in freshwater ecosystems [34,35,36].

Living organisms in freshwater ecosystems have been used for many years as sentinels against water pollution, especially organic pollution. For example, the responses of selected model species to different concentrations of various effluents have been the focus of toxicological analyses in controlled experiments [37]. Other programs have evaluated changes in populations or communities in natural waterways. Changes in populations or communities of fish, algae, and especially macroinvertebrates have been used for this purpose most often [38]. On average, species of mayflies (Ephemeroptera), stoneflies (Plecoptera), and caddisflies have been shown to be less tolerant of organic and other types of pollution than other freshwater taxa [38] and, therefore, changes in abundance and diversity in these three taxa are often emphasized in freshwater biomonitoring programs. Recently, evaluation of diverse functional traits of caddisflies and other freshwater species have provided richer insight into the probable causes of ecological degradation in surface waters [12] and even the effects of climate change [39,40,41].

Larvae and adults of caddisflies (along with mayflies, stoneflies, true flies, and a few other taxa) are commonly eaten by fish, crayfish, birds, bats, and other predators of human concern. Ecologically, the death of adults in riparian habitats and consumption of adults by terrestrial predators helps to assure a return of nutrients from freshwater ecosystems to the surrounding riparian environment [42]. The consumption of the immature stages by fish and crayfish in the water, helps assure that these resources are available as food for humans. Fish predation of caddisflies is the basis for sport fishing, especially the practice of tying imitations of caddisflies and other taxa on hooks to attract and snag feeding fish [43].

Another ecosystem service provided by net-spinning caddisflies is that they contribute to substrate stability by linking mineral substrates with their silk. Through their use of silk fastened to substrate sediment, these insect engineers prevent the mobilization of sediments and generally moderate transport of sediment, thereby reducing stress on benthic biota in storm events [44,45,46,47], and providing substrate stability even in ambient conditions [48,49,50,51]. Close to the substrate, in the “shadow” of caddisfly filter nets, other benthic macroinvertebrates find refuge in slower flows [52].

Caddisflies have also contributed to resolution of forensic questions, helping to determine time of death by their scavenging activity [53] or by inclusion of forensic evidence in their cases [54].

Much of the ecological diversity of caddisflies has been attributed to their use of silk [16,17]. The labial silk glands of caddisfly larvae are huge, extending posterad and filling much of the abdomen [55], enabling production of large amounts of proteinacous fibroin and sericin silk thread which is similar to that of Lepidoptera caterpillars. However, the silk produced by caddisfly larvae, unlike that produced by their lepidopteran cousins, adheres to objects underwater, e.g., [56]. This has led to exciting research into the properties of caddisfly silk, “a unique source of design principles for tough, self-healing synthetic materials, fibers, and fabrics, especially suited for underwater and wet-field biomaterial applications, as well as biomimetic processes for spinning water-borne polymers into fibers underwater” [57].

Caddisflies may cause negative impacts, however. For example, net-spinning caddisflies may block water intake pipes of hydro-electric power plants [58,59]. Sometimes, caddisflies destroy man-made wooden structures [60]. Some people are allergic to the hair of adult caddisflies [61]. Also, the number of emerging adult caddisflies near waterways can be so great that they are nuisances, irritating the people on whom they land, obscuring automobile windshields with their crushed bodies, or fouling wet paint [25,62]. Some species of Leptoceridae (species of *Triaenodes* and Triplectidinae) and Limnephilidae are pests of rice [26,28] and some species of Limnephilidae (e.g., *Limnephilus lunatus* Curtis, 1834, and *Drusus annulatus* (Stephens, 1837)) have been reported as pests of commercial water cress [28,63,64].

## 4. Phylogenetic Relationships of Trichoptera Families

Phylogenetic relationships among the Trichoptera taxa have been debated for more than 100 years, mostly from morphological data [65]. More recently, molecular studies have contributed to resolutions of lingering questions [66,67]. In particular, the relationships of certain basal lineages in the order have been especially challenging to ascertain. The families Glossosomatidae, Hydrobiosidae, Hydroptilidae, Ptilocoloepidae, and Rhyacophilidae have been variously classified with Annulipalpia or Integripalpia or in their own suborder “Spicipalpia,” the latter generally acknowledged as paraphyletic, but characterized by a distinctive semipermeable pupal cocoon [28]. The phylogenetic relationships inferred by Kjer et al. [67] are the most recently published, concluding that these five debated lineages constitute an ancient grade at the base of Integripalpia (Figure 2), similar to the arrangement inferred by Ross [68] 60 years earlier from morphological data. Here we refer to them informally as “Basal Lineages of Integripalpia.”

While the evolutionary history of caddisflies has been the subject of debate, with various research groups and analyses generating conflicting results, some consensus is being reached concerning the arrangement of families within suborders. In particular, the affinities of particular families with one another are becoming clearer as data sets become larger, based on hundreds or thousands of genes (Figure 1 and [69]). In suborder Annulipalpia, Hydropsychidae is consistently recovered as the first divergence and as sister to the rest of the suborder [66,67]. Philopotamidae and Stenopsychidae, together, then form the next diverging clade. The arrangement of the rest of the families in the suborder is the subject of some disagreement, but with clear sister relationships between Psychomyiidae and Xiphocentronidae and between Economidae and Polycentropodidae.

In suborder Integripalpia, it is becoming evident that the “spicipalpian” families (Hydroptilidae, Ptilocolepidae, Rhyacophilidae, Hydrobiosidae, and Glossosomatidae) and possibly the fossil family †Necrotauliidae form a grade (“Basal Lineages”) leading to the tube-case makers (Brevitentoria + Plenitentoria). We include †Necrotauliidae in Trichoptera-Integripalpia based on arguments by [75], tentatively accepted by [77], and especially on those by [83] who reported a synapomorphic haustellum in a †necrotauliid fossil from Early Jurassic-Sinemurian Age (mean 193.05 Ma).

Molecular data indicate that Hydroptilidae + Ptilocolepidae form the first branch of extant families in that grade, with the relationships of the remaining three extant families and the tube-case maker lineage ambiguous, illustrated as a quadrichotomy in Figure 1 [67,69,70]. Within the monophyletic tube-case-maker lineage there are two primary clades, infraorders Plenitentoria and Brevitentoria. In Plenitentoria, Phryganopsychidae is consistently recovered with molecular data as the first diverging lineage and as sister to the rest of the group. Relationships among the remaining families is somewhat ambiguous, although two clusters of families are consistently recovered: one clade giving rise to the families Apataniidae, Goeridae, Limnephilidae, Rossianidae, Thremmatidae, and Uenoidae (= Limnephiloidea [80]) and the other clade ancestral to Oeconesidae, Pisuliidae, and probably Kokiriidae. The relationships among families in Brevitentoria are less clear, with the families Atriplectidae, Limnocentropodidae, Tasimiidae, Calmoceratidae, Molannidae, Leptoceridae, Odontoceridae, and Philorheithridae forming a grade leading to a monophyletic Sericostomatoidea [82,84]. Within this grade, the families Calamoceratidae, Molannidae, and Leptoceridae form a clade (= Leptoceroidea sensu stricto), Limnocentropodidae forms a clade with Tasimiidae, and Odontoceridae forms a clade with Philorheithridae [67].

## 5. Evolution of Diversity and Ecosystem Services in Trichoptera

### 5.1. Annulipalpia Stationary Retreats and Feeding: Filtering-Collecting-Grazing-Gardening-Ambushing

Larvae of suborder Annulipalpia have been characterized generally as fixed retreat makers, building stationary shelters of silk and often bits of plant material and mineral particles. These retreats are generally attached to larger stones, woody debris, aquatic plants, exposed roots of riparian plants, or other relatively stable substrates, mostly in moving water such as streams or the wave-washed shores of large lakes. The primary apparent functions of these retreats, similar to the functions of portable cases, are physical shielding against predation—cloaking against detection by visual predators such as fish and sometimes from potential prey—and channeling oxygenated water past the body, usually from anterior to posterior, often with ventilating undulations of the body, although rheophilic larvae rely less on the latter function than do limnephilid larvae [27].

The stationary retreat of a larval hydropsychid (“net-spinning caddisfly”) typically is associated with a relatively flat filter net constructed beside the anterior, upstream entrance to the retreat (Figure 3). The retreat is constructed on the top or side of relatively stable substrate and may be a simple silk-and-plant-fragment tube or cornucopia with the net supported by retreat materials and exposed to the inflowing and outflowing current (e.g., *Arctopsyche* spp. [85]) or with the net protected by retreat materials. In the latter circumstance, the inflowing water may be constrained by a tubular foyer that may be elevated above the substrate and with a vertical or angular upstream pitot-tube opening and the outflowing water constrained by a downstream exit tube (e.g., *Macrostemum* spp. [86]). For either architectural arrangement, the net is spun transversely across the current, with mesh sizes varying among the different genera and species in correlation with current speeds and sizes of food particles typically consumed by those taxa [85,86,87,88].

Larvae of the next oldest extant lineage of Annulipalpia, the ancestor of the Philopotamidae (“finger-net caddisflies”) and Stenopsychidae, also filtered suspended organic particles from flowing water. Modern Philopotamidae larvae typically spin capture nets in slowly moving water, usually on the undersides of stones, with the smallest mesh openings recorded for Trichoptera [89]. The philopotamid net has no separate retreat, as a larva occupies all of a single sac-like or finger-like net. The net is attached to the substrate only at its anterior, upstream end, often with small pebbles around this entrance to help keep it open. The remainder of the net is unrestrained and wafting gently in the current with a smaller opening at the posterior end (Figure 4) [89]. Stenopsychidae make loosely woven retreats/capture nets in fast-flowing water, with three different parts of the capture-net portion serving as: (1) a cover, (2) a primary feeding mesh, and (3) a pocket for collecting seston (minute living and nonliving FPOM suspended in moving water); the retreat portion is beside the latter [90].

In superfamily Psychomyioidea, stationary-retreat-making larvae feed by filtering suspended FPOM (Dipseudopsidae and some Polycentropodidae), collecting deposited FPOM (Xiphocentronidae and some Psychomyiidae), scraping or grazing on periphyton (biofilm) (most Psychomyiidae), farming algae in the walls of the retreat (some Psychomyiidae), or preying on small invertebrates (Ecnomidae and most Polycentropodidae). The filter net of a larva of Dipseudopsidae (“pitot-tube caddisfly”) is constructed across the outflow branch of a mine burrowed in fine sediment, with the walls of the mine reinforced with silk [91]. Larvae of sister families Xiphocentronidae and Psychomyiidae (“net-tube caddisflies”) construct “lairs” that hide them from potential predators as they feed. The larva of at least one species of *Tinodes* (Psychomyiidae) gardens diatoms in its silken lair, ingesting its older portions and their cultivated diatoms while extending the newer end to protect it while it grazes on algae on the substrate [92,93,94]. In Polycentropodidae (“trumpet-net caddisflies”), larvae of *Neureclipsis* and *Polycentropus* spin bag-like filtering nets in slowly flowing water to filter FPOM and to capture small invertebrate prey; a larva fastens the upstream ends to stable substrate such as aquatic plants or submerged tree roots and branches with the downstream middle gently wafting freely in the slowly flowing current while a large anterior opening directs the flowing water into the net and through the downstream wall of the net (Figure 5) [95]. Larvae of polycentropodid genera *Holocentropus* and *Plectrocnemia* (Figure 6a) are predators with silken retreats suspended on aquatic plants and with many irregularly radiating silken threads to form capture nets [95] functioning much like those of terrestrial web-spinning spiders (Figure 6b). The flattened lairs of predatory larvae of genera *Cernotina*, *Cyrnellus*, and *Nyctiophylax* (Polycentropodidae) serve to hide them from both potential predators and prey. These lairs are often equipped with silken trip lines radiating from the two ends to alert a larva of potential intruders, allowing it to pounce from its lair, ambushing its unsuspecting prey [28].

Thus, in Annulipalpia, the current phylogeny supports the hypothesis that the architectural behavior of larvae has evolved from retreats with accompanying filter nets to retreats themselves serving as filter nets to retreats adapted for predation, collecting FPOM, scraping periphyton, and gardening.

For pupation in Annulipalpia, the last instar larva constructs adjacent to the retreat a separate, fixed, dome-like shelter of silk and fine minerals. A cocoon is spun inside the shelter—with its walls either incorporated into the silk of the shelter and adhering to the underlying substrate or not. The cocoon is permeable, allowing oxygenated water to flow through it over the body of the larva and the pupa into which it transforms [28].

### 5.2. Integripalpia Basal-Lineage Case Making and Feeding: From Caselessness to Living under Domes and Collecting-Grazing-Piercing-Parasitizing-Pursuing

Larvae of Hydroptilidae and Ptilocolepidae (collectively “microcaddisflies”) are free-living and campodeiform (slender and with tapered ends) in the first four instars, but feed usually by gathering deposited FPOM. They exhibit hypermetamorphosis, completing those early instars rapidly and then transforming into the final larval instar with a much-enlarged abdomen that is generally flattened from side to side (compressed) or from top to bottom (depressed). This final instar larva constructs a silken case with its shape and any plant or mineral inclusions typical for the genus. This case may be fixed or portable and, if portable, usually is composed of two sheets of material, or valves, joined on the long edges and open at the ends, resembling a coin purse, hence another English common name, “purse-case-making caddisflies” (Figure 7). The fixed case in genera such as *Leucotrichia* or *Zumatrichia* consists of a single dorsal sheet attached laterally to the substrate [27]. The final instar of a microcaddisfly larva usually feeds in a genus-specific manner, by shredding living algae or liverworts, gathering deposited FPOM, scraping periphyton, or piercing and sucking contents of individual cells of filamentous algae [28]. Some species are parasitic [96] and a few entrain suspended FPOM in silken shelters spun over their cases [97]. Pupation occurs between these sheets or beneath the single sheet where the cocoon spun by the last instar larva in preparation for pupation is semipermeable in Hydroptilidae, but porous in Ptilocolepidae [28].

Larvae of Glossossomatidae (“saddle-case caddisflies”) are cyphosomatic scrapers of periphyton growing on the tops of rocks (Figure 8a). Each larval instar of most species constructs a dome-like case of stones and silk that is oval and has a transverse strap ventrally connecting the two longer sides (Figure 8b) [27,28]. Variations of this architecture have been observed among some species of the glossosomatid subfamily Agapetinae [98,99,100]. The ventral openings beyond the transverse edges of the strap allow the head, thoracic legs, and the end of the abdomen to engage the rock substrate. These openings are interchangeable, allowing the larva to reverse direction under the dome. When preparing for pupation, the final larval instar removes the transverse ventral strap, cements the dome to the rock substrate with silk, spins a semipermeable cocoon, and pupates inside it [28].

Larvae of Rhyacophilidae and Hydrobiosidae (collectively “free-living caddisflies”) are free-living and campodeiform, usually feeding as predators that pursue their prey (Figure 9). Their last larval instar assembles a fixed, dome-like shelter of stones and silk, under which it spins a semipermeable silken cocoon and pupates within it [28,101]. In this way, the pupal shelters and cocoons of Rhyacophilidae and Hydrobiosidae resemble those of Glossosomatidae [28].

Although the feeding strategies and body forms of the larvae in these extant integripalpian basal lineages are mostly different, at least two common themes should be mentioned. (1) Pupation usually occurs in a semipermeable cocoon. Oxygenated water diffuses by osmotic pressure through the cocoon wall, with concentrated organic molting fluid inside the cocoon establishing an osmotic pump [28,101]. (2) Pupation occurs under a dome-like shelter or, in Hydroptilidae and Ptilocolepidae, a modified version of a dome. The two sheets of the usual hydroptilid-ptilocolepid case can be compared with the two parts of the glossosomatid case (dorsal dome and ventral strap) fused along their edges. Even for a hydroptilid with a compressed larval abdomen, the pupa becomes depressed and is repositioned between the two sheets, similar to the orientation of a glossosomatid pupa with respect to its dome and the substrate [28]. In other words, the larvae of Hydroptilidae, Ptilocolepidae, and Glossosomatidae can be inferred to have constructed dome-like pupation shelters resembling those of Rhyacophilidae and Hydrobiosidae, but to have done so precociously in all larval instars (Glossosomatidae) or in the final instar only (Hydroptilidae and Ptilocolepidae). Thus, although the feeding strategies among these basal lineages are quite different, the evolutionary similarity of their architectural and physiological characteristics is striking.

### 5.3. Integripalpia: Tubular Cases and Feeding

Larvae of all Integripalpia in infraorders Plenitentoria (Figure 10) and Brevitentoria (Figure 11) construct portable cases that are essentially tubular in design, usually with discernible anterior and posterior ends. The cases are of a wide variety of shapes and are constructed with different materials and in different arrangements that are generally distinctive for the taxon. They live in such cases throughout their larval existence, usually adding case material to the gradually enlarging anterior end and usually cutting away and discarding case material from the narrower posterior end as they grow. The posterior end of the case commonly has a silken membranous sieve to protect the larva from intruders but allowing the flow of water through the case for respiration [28].

The many different ways that the tube-case-making Integripalpia feed are indicated in Table 1. Shredding detritivory is common, as many species, especially in Plenitentoria, consume mostly the same kinds of leaves or wood with which they construct their cases [102,103,104]. Larvae of some species of Plenitentoria also eat animal tissue, shredding and ingesting living insects and vertebrate eggs as predators (e.g., Phryganeidae: *Banksiola*, *Oligostomis*, *Oligotricha*, *Phryganea*, *Ptilostomis*; Lepidostomatidae: *Lepidostoma* [21,27,28,105,106]) or cannibals (e.g., Limnephilidae: *Asynarchus* [107]) or scavengers (e.g., Lepidostomatidae: *Lepidostoma*; Limnephilidae: *Dicosmoecus*, *Psychoglypha* [108]). An interesting parallel development in Integripalpia is filter feeding; whereas Annulipalpia larvae commonly use silken filter nets to remove FPOM from suspension, larvae of some Plenitentoria, such as *Brachycentrus* spp. (Brachycentridae) and *Allogamus* spp. (Limnephilidae), filter suspended FPOM with hairs on their legs (Figure 12).

In the integripalpian infraorder Brevitentoria, most larvae construct tubular cases of mineral materials, occasionally with plant parts attached [27,28,109]. Exceptions constructing cases primarily of plant material include Calamoceratidae (*Anisocentropus and Heteroplectron*), Leptoceridae (*Triaenodes*), Calocidae (*Caenota*), Helicophidae (some *Alloecella*), Oeconesidae, and Conoesucidae (some *Conoesucus and* some *Costora*) [27,28,109,110,111]. Larvae of several families of Brevitentoria are essentially omnivorous [1,27,28], but some interesting specializations have been observed. Larvae of Atriplectididae are scavengers with long heads and telescoping mesothoraces, enabling them to pierce the exoskeletons of dead arthropods to eat the soft tissue inside [112]. Like annulipalpian *Holocentropus* and *Plectrocnemia* species, Limnocentropodidae are filtering predators, but instead use their spiny legs to capture insect larvae similar to the way *Brachycentrus* and *Allogamus* filter FPOM; each has its case suspended in turbulent current by a long silken stalk anchored to a boulder [113]. Larvae of some species of *Ceraclea* (Leptoceridae) prey exclusively or facultatively on freshwater sponges [114,115] or freshwater snails [116]. Larvae of *Pseudogoera* (Odontoceridae) hang in mosses of waterfalls, preying on small insects [27,117]. Larvae of families Antipodoeciidae, Helicopsychidae, Petrothrincidae, and Tasimiidae feed mainly as grazers [28]. Among shredding herbivores, larvae of Chathamiidae are marine commensals of starfish in intertidal rock pools, constructing cases of coral or coralline algae and apparently consuming the red alga *Corallina* [27,109,118]. Larvae of Helicophidae shred and consume detritus, mosses, and liverworts [119] and possibly also graze on attached periphyton (algae, fungi, microbes, and detritus adhering to underwater surfaces) [28].

In preparation for pupation, integripalpian larvae generally seal both ends of the larval case with a silken sieve. They then pupate inside the modified larval case, shedding the exuviae of the final larval instar and usually depositing them at the posterior end of the case [28]. Except for the fact that integripalpian larvae do not spin cocoons, this pupation behavior resembles the behavior of larval Hydroptilidae, Ptilocolepidae, and Glossosomatidae, suggesting that the larval case of Integripalpia is also a pupation shelter that is constructed by the larva precociously for use by all larval instars.

### 5.4. Evolution of Angiosperm Phytophagy and Case Material

In their molecular analyses, Malm et al. [66] estimated the age of Trichoptera as “around 234 Ma, i.e., Middle–Late Triassic.” Unpublished molecular data indicate an earlier origin of about 262 Ma (Middle Permian, Guadalupian Epoch, Capitanian Age) [70]. The oldest known trichopteran fossils are in the extinct integripalpian family †Necrotauliidae: †*Necrotaulius* †*proximus* Sukatsheva, 1973, and †*Prorhyachophila* †*furcata* Sukatsheva, 1973, both described from Middle–Late Triassic [77,83]. Subsequently, these fossils were placed in the Late Triassic-Carnian Age (222.6 Ma) [6]. The orders Trichoptera and Lepidoptera are considered sister lineages that arose from among several other, now extinct, lineages in superorder Amphiesmenoptera [120]. For comparison, the oldest undisputed lepidopteran fossils date from ca. 190 Ma, in Early Jurassic [121] or ca. 210 Ma, in Late Triassic [122].

The plant material consumed by larvae of most Annulipalpia, Hydroptilidae, Ptilocolepidae, and Glossosomatidae is non-vascular (algae, diatoms, liverworts). Although there are many examples of larval caddisflies using non-vascular plants in cases and as food [119], the plant material used by most modern plenitentorian larvae is usually leaves or wood of angiosperms, not gymnosperms or non-vascular plants. Assuming that the diet of plenitentorian larvae has not changed since their origin, this observation suggests that angiosperms evolved at least by the time that Plenitentoria appeared. Unpublished molecular data [69,70] show that Plenitentoria diverged from a common ancestor with Brevitentoria about 175 Ma in the Early Jurassic-Toarcian Age (95% confidence intervals 192.21–153.65 Ma). These results agree with the oldest known fossil species of Plenitentoria, †*Dysoneura* †*trifurcata* Sukatsheva, 1968, a species of †Dysoneuridae, also about 175 Ma, from the Early Jurassic-Toarcian Age [83]. The oldest extant lineage in Plenitentoria for which we have molecular sequences is Phryganopsychidae. That family diverged from the remaining Plenitentoria families for which we have molecular data about 144 Ma (95% confidence intervals 161.1–128.46 Ma) [69,70].

There are 11 ichnogenera and 245 ichnospecies of Trichoptera. These are parataxa which are formally named for cast impressions of caddisfly larval cases, i.e., the “fossilized work of an organism”, for which names are regulated by the International Code of Zoological Nomenclature [123]. In trichopterology, these names are not assigned to any ichnofamily nor usually to any fossil or extant family. Overviews of Trichoptera ichnotaxa were provided by [79,120,124,125]. Most ichnospecies date from Cretaceous to Miocene, but a few are known from Jurassic deposits, including †*Scyphindusia* †*hydroptiliformis* Sukatsheva, 1985, of Hydroptilidae (possibly †Vitimotauliidae [125]), from the Early Jurassic-Middle Jurassic [126]. Other than the case of that ichnospecies, the cases of caddisfly ichnospecies are tubular, conforming to the shape of modern cases of Plenitentoria and Brevitentoria. Ichnogenera of Trichoptera are named according to the type of material incorporated into their cases. Among these, the cases of only the ichnogenus †*Folindusia* are composed of vascular plant material, indicating that they may be larval cases of Plenitentoria. For many years, the oldest known †*Folindusia* have been five ichnospecies from the Middle Jurassic [127]. A recent report cited unnamed †*Folindusia* species from the Middle Triassic-Late Ladinian Age, 238–237 Ma [128]. Fossil caddisfly cases reported from Early Permian marine deposits [129] are inconsistent with that fossil record and known caddisfly biology and have been dismissed as probable work of worms [128].

Therefore, our molecular findings and the fossil record in Trichoptera suggest that the angiosperms evolved by at least 175 Ma, when probable-angiosperm-consuming caddisflies appeared. Recent estimates from plant molecular studies tend to agree with our results, for example suggesting an angiosperm origin in Late Triassic [130] or earlier [131]. Those molecular botanical findings seem to be confirmed by recently described fossils of *Nanjinganthus dendrostyla* Fu et al., 2018, a noncarpellate epigynous flower from 174 Ma, in Early Jurassic [132,133], very similar to our estimated age for the origin of Plenitentoria based on molecular data and not inconsistent with the fossil evidence of Middle Jurassic plenitentorians and ichnotaxa.

Much work remains to decipher the phylogenetic history of Trichoptera. However, nascent efforts, such as the Trichoptera 1KITE subproject [134] and mitochondrial genome projects [135] that are focused on resolving the Trichoptera tree of life using large data sets gathered from modern DNA sequencing methods, such as genomes [136], transcriptomes, and targeted exon capture, are poised to resolve many of the contentious splits within the Trichoptera tree. Once enabled with well-resolved phylogenies, researchers will be better able to interpret the foundation of functional traits and related ecological services within Trichoptera and predict those that remain unknown, thereby focusing and directing investigations to observe them.

## 6. Conclusions

The species diversity of Trichoptera is greater than that of all other primarily aquatic insect orders combined, rivaling that of aquatic beetles and equaling about one-third of that of aquatic flies. The diversity of trophic relationships is also greater than that of any other order of aquatic insects other than Diptera. Other functional traits are unusually diverse, as well, contributing to the diversity of ecological services they perform. In light of the much greater number of Diptera species in freshwater habitats, the relative diversity of caddisfly functional traits appears greater than that of all other freshwater insects and, indeed, probably is greater than of all other freshwater macroinvertebrates. Together with consideration of their ubiquitous presence and of their density in relatively unpolluted waters, this ecological diversity suggests that caddisflies are vital for the sustainable function of most natural freshwater ecosystems. The current and potential value of Trichoptera for human interests also is considerable. Their occasional negative impacts can be of concern, as well.

Knowledge concerning Trichoptera is still sorely needed. The rate of species discovery is very high and even accelerating in the Oriental and Neotropical Biogeographical Regions. New molecular research tools also are contributing to our ability to uncover cryptic species and to associate life stages (e.g., larvae and other life stages of many species are currently unidentifiable and need to be associated with their identifiable adults, mainly males). These latter associations make possible the discovery and description of diagnostic characters for identifying benthic forms. As larvae and other life stages of more species become identifiable, possibilities increase substantially for studying their functional traits, ecological services, life histories, and habitat requirements and for using them in biomonitoring programs. At the same time, species of Trichoptera are becoming extinct at an alarming rate. Therefore, because of the importance of Trichoptera in freshwater ecosystems, the urgency to increase our knowledge of the diversity and ecosystem services of caddisflies is great and increasing. Each of the authors of this review is eager to accept capable and enthusiastic young scholars into our laboratories to investigate these topics.

The phylogenetic pattern for Trichoptera is gradually becoming clearer. It reveals two major clades that have each diversified extensively for more than 200 million years. This evolutionary success seems to have resulted mainly from the versatility of silk and the ability of larvae to manufacture and dispense it efficiently under water. Larvae in successive lineages have used silk successfully in different types of stationary retreats and portable cases to protect themselves from predators visually and physically and to enhance their respiration efficiency. They have used silk also to filter various nutrient particles from suspension, to anchor themselves from dislodgement in moving water, to line and reinforce mines in fine sediment, and to serve as capture nets for their prey. The phylogeny also suggests hypotheses regarding structural and functional characteristics of relatives for which these characteristics have not yet been observed. The many variations on ways caddisfly larvae access nutrients and make them available to other organisms in the freshwater trophic network and the ways that their silken structures engineer their habitats make them especially important for the structure and function of their ecosystems.

The phylogenetic pattern also reveals the possible evolutionary origin of angiosperm consumption and use in case construction, especially by larvae in the integripalpian infraorder Plenitentoria. This insight may contribute to the resolution of questions about the age of this major plant lineage, implying that angiosperms arose by 175 Ma.

## Figures and Tables

**Figure 1 insects-10-00125-f001:**
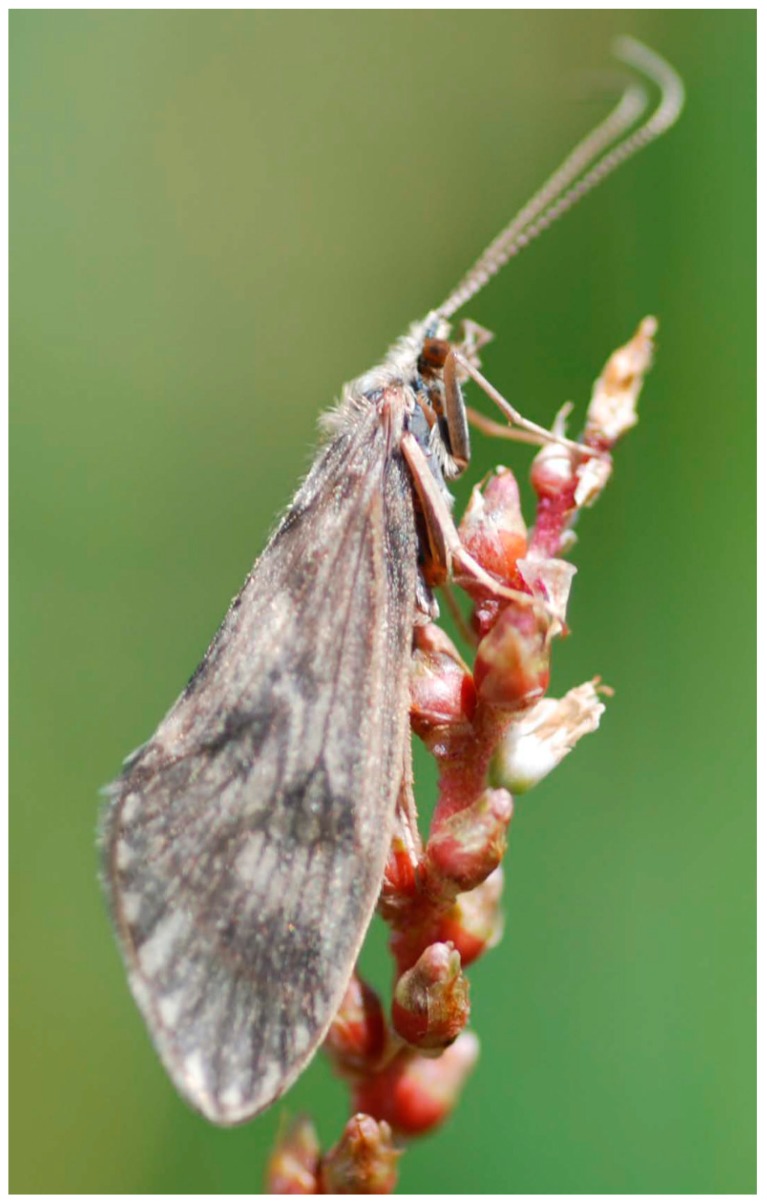
Adult of *Brachycentrus americanus* (Banks, 1899) (Brachycentridae). © J.C. Morse.

**Figure 2 insects-10-00125-f002:**
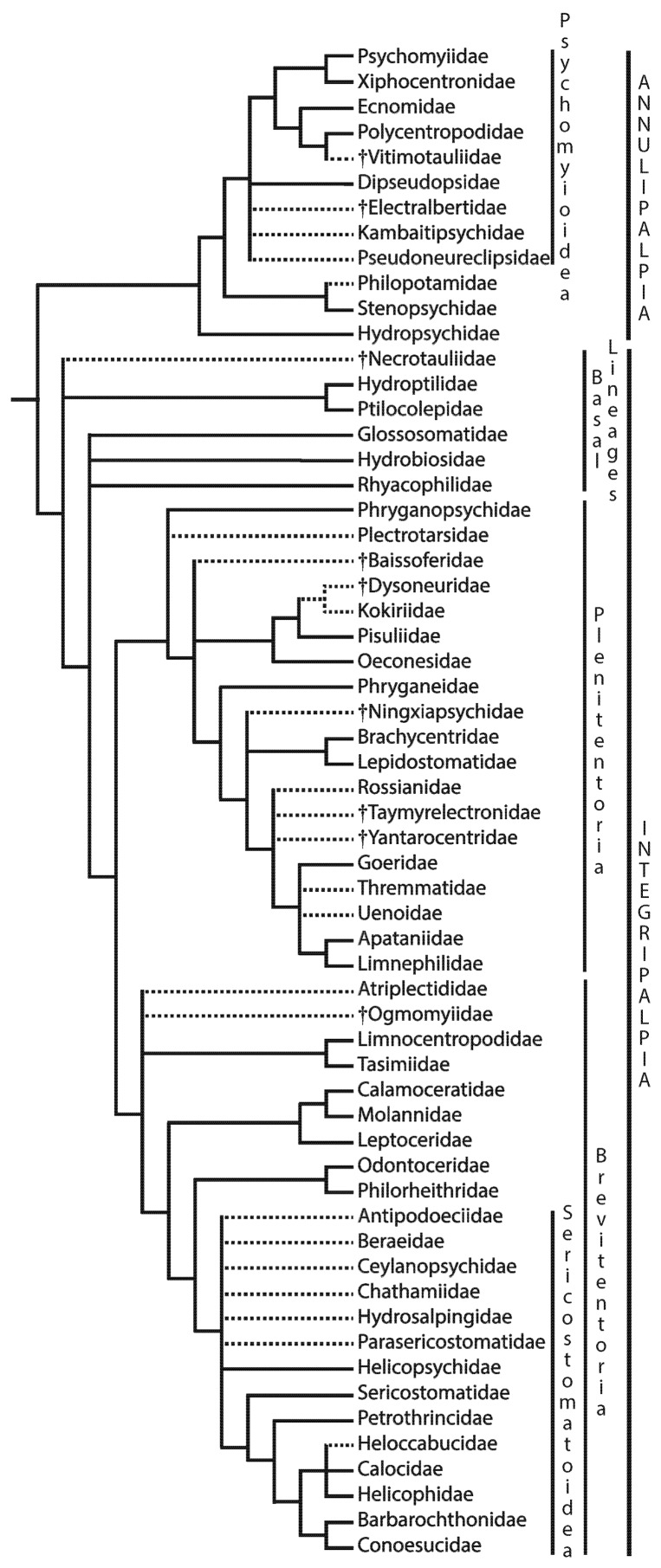
Phylogeny of extinct (†) and extant Trichoptera families. Solid lines refer to relationships inferred with molecular data by Kjer et al. [67] with some refinements [69,70]. Dashed lines refer to relationships inferred by other data (mostly morphological) from various sources: †Vitimotauliidae [71,72], Dipseudopsidae [73], †Electralbertidae [74], Kambaitipsychidae [75], Pseudoneureclipsidae [75], Philopotamidae [73], †Necrotauliidae [76,77], Plectrotarsidae [73], †Baissoferidae [72], †Dysoneuridae [78], Kokiriidae [73], †Ningxiapsychidae [79], Rossianidae [73], †Taymyrelectronidae [80], †Yantarocentridae [8], Thremmatidae [81], Uenoidae [73], Atriplectididae [73], †Ogmomyiidae [9], Ceylanopsychidae [82], Antipodoeciidae [82], Beraeidae [82], Chathamiidae [73], Hydrosalpingidae [82], Parasericostomatidae [82], and Heloccabucidae [82]. Lengths of lines are arbitrary and uninformative.

**Figure 3 insects-10-00125-f003:**
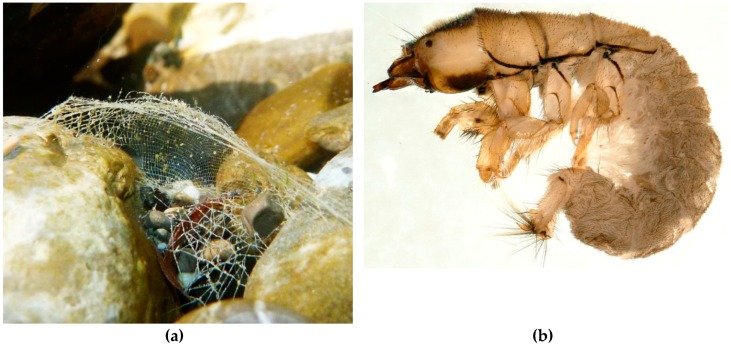
(**a**): Filter net and gravel retreat of *Hydropsyche* sp. (Hydropsychidae) between large stones. (**b**): Larva of *Hydropsyche bulgaromanorum* Malicky, 1977 (Hydropsychidae). Both images © W. Graf.

**Figure 4 insects-10-00125-f004:**
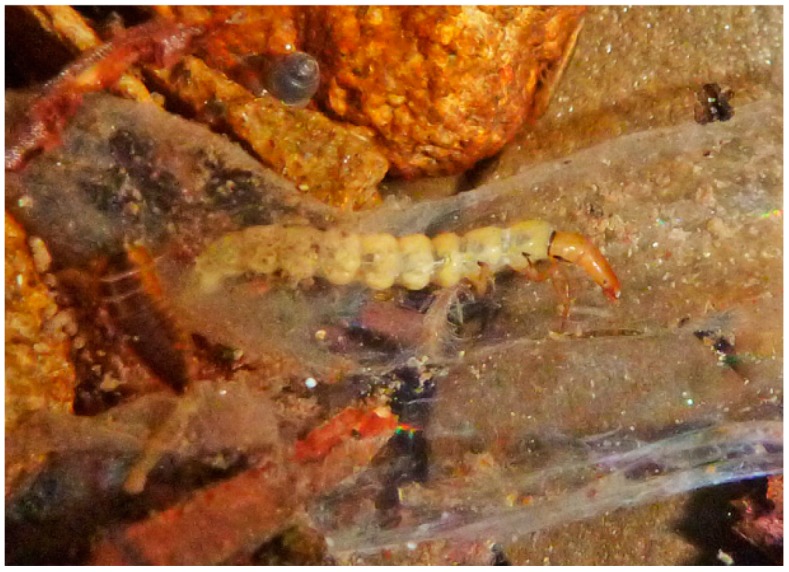
Larva of *Wormaldia* sp. (Philopotamidae) in its filter net. © W. Graf.

**Figure 5 insects-10-00125-f005:**
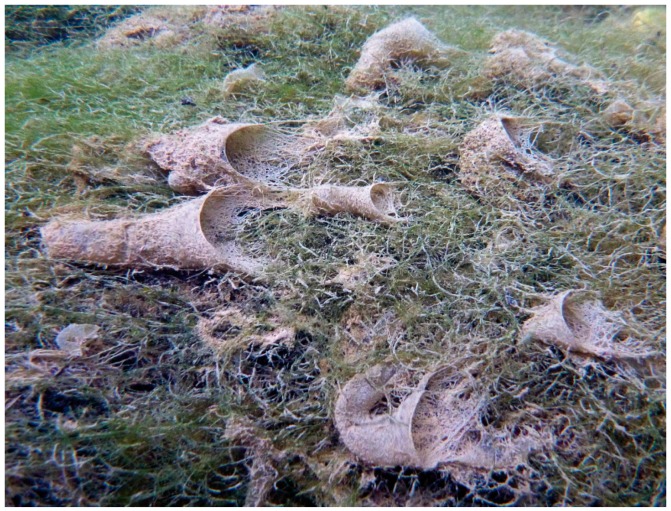
Capture nets of *Neureclipsis* sp. (Polycentropodidae). © W. Graf.

**Figure 6 insects-10-00125-f006:**
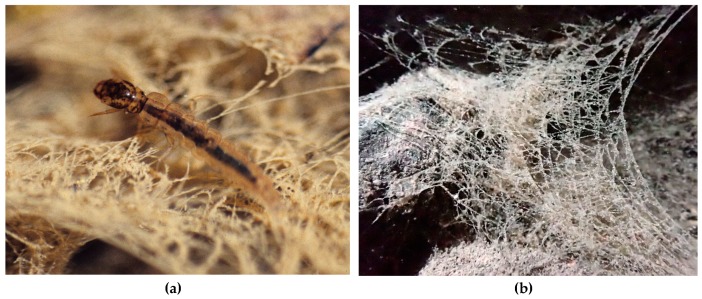
(**a**): Larva of *Plectrocnemia conspersa* (Curtis, 1834) (Polycentropodidae). (**b**): Capture net of *P. conspersa*. Both figures © W. Graf.

**Figure 7 insects-10-00125-f007:**
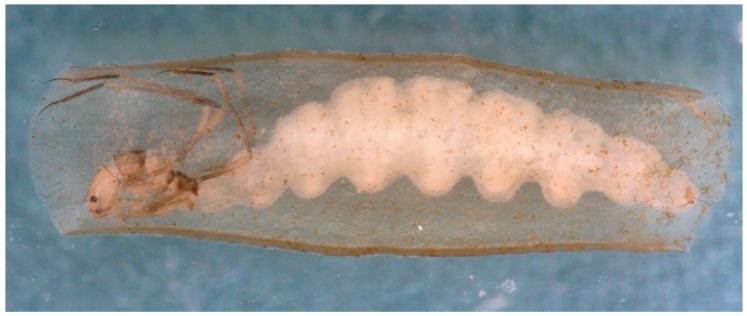
Larvae of *Tricholeiochiton fagesii* (Guinard, 1879) (Hydroptilidae) in its case. © W. Graf.

**Figure 8 insects-10-00125-f008:**
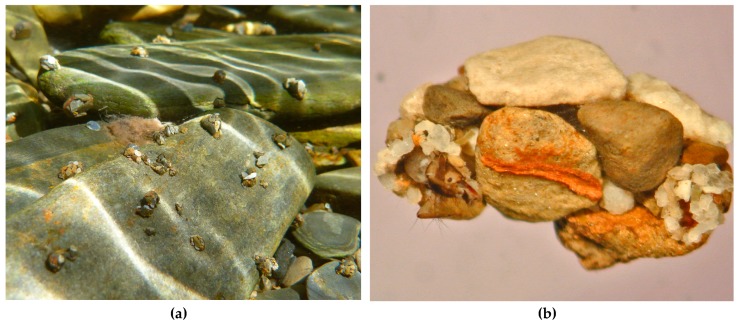
(**a**): Larvae of Glossosomatidae grazing epilithic periphyton from large stones. (**b**): Larva of *Synagapetus* sp. (Glossosomatidae) in its case. Both figures © W. Graf.

**Figure 9 insects-10-00125-f009:**
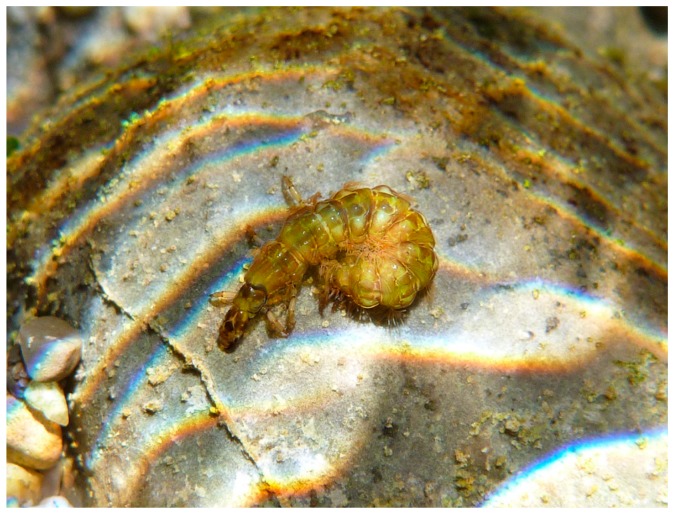
Larva of *Rhyacophila* sp. (Rhyacophilidae). © W. Graf.

**Figure 10 insects-10-00125-f010:**
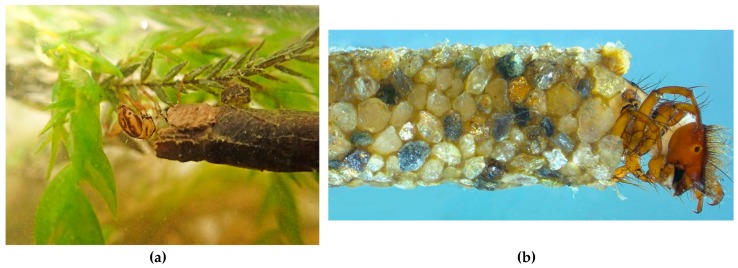
(**a**): Larva of *Oligostomis reticulata* (Linnaeus, 1761) (Plenitentoria: Phryganeidae) in its case composed of pieces of angiosperm leaves; © W. Graf. (**b**): Larva of *Drusus* sp. (Plenitentoria: Limnephilidae) in its case composed of gravel; © S. Vitecek.

**Figure 11 insects-10-00125-f011:**
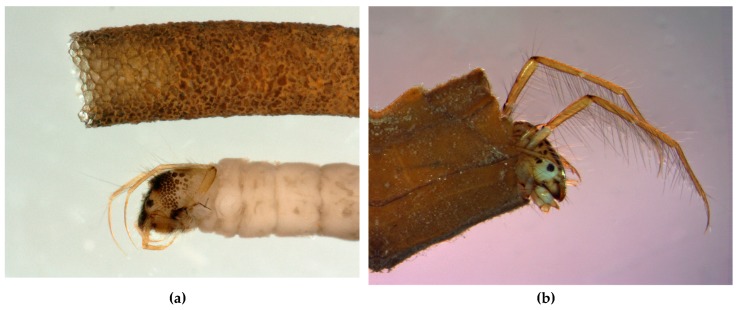
(**a**): Larva of *Beraeodes minutus* (Linnaeus, 1761) (Brevitentoria: Beraeidae) and its case composed of fine sand. (**b**): Larva of *Triaenodes bicolor* (Curtis, 1834) (Brevitentoria: Leptoceridae) in its case composed of pieces of angiosperm leaves. The rows of long hairs on the legs are used for swimming. Both figures © W. Graf.

**Figure 12 insects-10-00125-f012:**
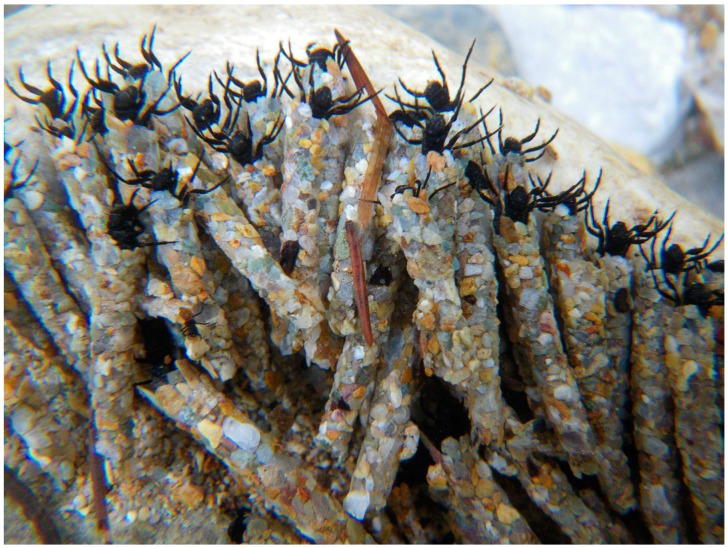
Larvae of *Allogamus auricollis* (F.J. Pictet, 1834) (Plenitentoria: Limnephilidae) filtering FPOM with their hairy legs. © W. Graf.

**Table 1 insects-10-00125-t001:** Extant and fossil (†) families of Trichoptera, each with earliest fossil epochs, mean geological age, larval cases or retreats, feeding types, numbers of extant and extinct genera and species of the world, and numbers of extant species in seven biogeographic regions.

Families	Earliest Fossils ^1^	Mean Geological Age (Ma) ^2^	Cases/ Retreats ^3^	Feeding ^4^	Number of Extant Genera ^5^	Number of Fossil Genera ^5^	Number of Extant Species ^5^	Number of Fossil Species ^5^	AT Species ^5^	AU Species ^5^	EP Species ^5^	NA Species ^5^	NT Species ^5^	OL Species ^5^	WP Species ^5^
Psychomyiidae	K_2_	77.05	rt	gra,xyl, gat,pff	10	3	600	18	43	12	44	18	0	378	116
Xiphocentronidae	MI	21.73	rt,tm	gat	7	0	183	1	2	0	3	8	58	116	1
Ecnomidae	EO	35.55	rt	pre,pff	12	1	518	22	98	183	4	3	57	169	11
Polycentropodidae	K_1_	142.85	rt,ca	pre,pff	18	11	861	117	21	86	53	9	282	295	72
†Vitimotauliidae	J_3_	148.15	N/A	N/A	0	7	0	24	0	0	0	0	0	0	0
Dipseudopsidae	K_2_	91.1	rt	pff	5	1	112	10	51	4	3	6	0	53	2
†Electralbertidae	K_2_	77.05	N/A	N/A	0	1	0	1	0	0	0	0	0	0	0
Kambaitipsychidae	N/A	N/A	N/A	N/A	1	0	2	0	0	0	0	0	0	2	0
Pseudoneureclipsidae	N/A	N/A	rt	gat?	2	0	121	4	5	1	3	0	4	99	9
Philopotamidae	J_1_	179.3	fn	pff	26	10	1508	35	150	215	52	76	354	625	65
Stenopsychidae	EO	35.55	rt,fn	pff	3	0	107	1	1	9	26	0	3	83	0
Hydropsychidae	EO	35.55	rt,fn	pff,pre, gra,gat	41	2	1982	12	221	181	89	173	457	757	171
†Necrotauliidae	TR_3_	222.6	N/A	N/A	0	9	0	28	0	0	0	0	0	0	0
Hydroptilidae	K_1_	118.5	fl,pc	pie,gra, gat,pre, shh,par	74	1	2570	22	161	337	89	352	921	614	183
Ptilocolepidae	K_2_	77.05	fl,pc	shh	2	0	16	2	0	0	7	2	0	2	5
Glossosomatidae	J_3_	148.15	ps	gra	20	1	785	10	6	40	73	100	318	181	91
Hydrobiosidae	J_3_	148.15	fl	pre	51	5	427	6	0	199	4	7	184	38	1
Rhyacophilidae	J_3_	158.4	fl	pre,shd, gra	5	3	833	14	0	0	122	130	0	469	133
Phryganopsychidae	N/A	N/A	pl	shd	1	0	4	0	0	0	2	0	0	3	0
Plectrotarsidae	J_3_	148.15	pl	shd	3	1	5	1	0	5	0	0	0	0	0
†Baissoferidae	J_3_	148.15	N/A	N/A	0	1	0	5	0	0	0	0	0	0	0
†Dysoneuridae	J_2_	173.6	N/A	N/A	0	75	0	14	0	0	0	0	0	0	0
Kokiriidae	N/A	N/A	mn	pre	6	0	15	0	0	13	0	0	2	0	0
Pisuliidae	N/A	N/A	pl	shd	2	0	19	0	19	0	0	0	0	0	0
Oeconesidae	N/A	N/A	pl,mn	shd	6	0	18	0	0	18	0	0	0	0	0
Phryganeidae	J_3_	148.15	pl	pre,shh, gra,gat	15	7	81	39	0	0	39	28	0	22	18
†Ningxiapsychidae	K_1_	105.8	N/A	N/A	0	1	0	1	0	0	0	0	0	0	0
Brachycentridae	K_1_	137.05	pl,mn,si	gra,pff, shh,pre, gat	7	1	112	2	0	0	30	39	0	31	28
Lepidostomatidae	K_1_	127.5	pl,mn	xyl,shd, gra,pre	7	4	519	14	49	3	74	77	22	287	31
Rossianidae	N/A	N/A	mn	gat,gra, shh	2	0	2	0	0	0	0	2	0	0	0
†Taymyrelectronidae	K_2_	84.65	N/A	N/A	0	1	0	1	0	0	0	0	0	0	0
†Yantarocentridae	EO	46.5	N/A	N/A	0	1	0	1	0	0	0	0	0	0	0
Goeridae	EO	35.55	mn	gra,gat	11	0	189	4	2	5	30	12	0	125	18
Thremmatidae	N/A	N/A	mn	gra,gat	3	0	53	0	0	0	9	36	0	5	5
Uenoidae	N/A	N/A	mn	gra,gat	4	0	32	0	0	0	1	20	0	11	0
Apataniidae	N/A	N/A	mn	gra,gat, shd,shh	21	1	212	1	0	0	83	32	0	70	41
Limnephilidae	EO	35.55	mn,pl	shd,gra, pre,gat, pff,shh	97	3	1037	12	0	2	198	251	43	142	557
Atriplectididae	N/A	N/A	mn	sca	4	0	6	0	2	3	0	0	1	0	0
†Ogmomyiidae	EO	46.5	N/A	N/A	0	1	0	3	0	0	0	0	0	0	0
Limnocentropodidae	N/A	N/A	mn	pre	1	0	18	0	0	0	2	0	0	17	0
Tasimiidae	N/A	N/A	mn	gra,gat	4	0	9	0	0	7	0	0	2	0	0
Calamoceratidae	J_3_	148..05	pl	shd,gra, pre	8	4	190	7	6	34	8	5	76	67	2
Molannidae	N/A	N/A	mn	pre,gat, gra	2	0	42	6	0	0	10	7	0	26	6
Leptoceridae	K_1_	137.05	mn,pl,si	shh,pre, gra,gat	49	9	2235	30	353	324	123	130	286	987	125
Odontoceridae	K_2_	84.65	mn	gra,gat, pre	15	4	172	8	4	5	14	15	47	89	3
Philorheithridae	N/A	N/A	mn	pre	9	0	30	0	3	21	0	0	6	0	0
Ceylanopsychidae	N/A	N/A	N/A	N/A	1	0	7	0	0	0	0	0	0	7	0
Antipodoeciidae	N/A	N/A	mn	gra	3	0	34	0	0	1	0	0	33	0	0
Beraeidae	EO	35.55	mn	gat,shd, gra	7	0	59	3	2	0	1	3	0	0	53
Chathamiidae	N/A	N/A	cl	shh	2	0	5	0	0	5	0	0	0	0	0
Hydrosalpingidae	N/A	N/A	si	gat,gra	1	0	1	0	1	0	0	0	0	0	0
Parasericostomatidae	N/A	N/A	N/A	N/A	2	0	13	0	0	0	0	0	13	0	0
Helicopsychidae	EO	35.55	mn	gra	2	5	281	15	17	53	2	14	116	80	5
Sericostomatidae	K_2_	84.65	mn,si	shd,gat, shh,pre	16	4	103	6	19	0	2	16	7	4	58
Petrothrincidae	N/A	N/A	mn	gra	1	0	14	0	14	0	0	0	0	0	0
Heloccabucidae	N/A	N/A	N/A	N/A	1	0	1	0	0	1	0	0	0	0	0
Calocidae	N/A	N/A	mn,pl	gat	7	0	33	0	0	33	0	0	0	0	0
Helicophidae	K_1_	127.5	mn,pl,si	gat,shh	9	0	47	0	0	31	0	0	16	0	0
Barbarochthonidae	N/A	N/A	si,mn	shd	1	0	1	0	1	0	0	0	0	0	0
Conoesucidae	N/A	N/A	mn,pl,si	gra,shd,shh	11	0	42	0	0	42	0	0	0	0	0
ichnotaxa	J_1_/J_2_	174.1	mn,pc,pl	N/A	0	11	0	265	0	0	226	10	1	0	31
TOTALS					618	121	16,266	765	1251	1873	1426	1581	3309	5854	1841

^1^ Periods and Epochs from [6], except Ptilocolepidae [7], †Yantarocentridae [8], †Ogmomyiidae [9]. Abbreviations: Mesozoic Periods: TR_2_ = Middle Triassic, TR_3_ = Late Triassic, J_1_ = Early Jurassic, J_2_ = Middle Jurassic, J_3_ = Late Jurassic, K_1_ = Early Cretaceous, K_2_ = Late Cretaceous; Cenozoic Epochs: EP = Paleocene, EO = Eocene, OL = Oligocene, MI = Miocene; N/A = fossils not available. ^2^ [10]. ^3^ [1,2]. ^4^ [1,2,11,12]. Abbreviations: gat = gatherers/scrapers of sedimented FPOM; gra = grazers/scrapers of endolithic and epilithic algal tissues, biofilm, partially FPOM, partially tissues of living plants; par = parasites; pff = passive filter feeders of suspended FPOM, CPOM, and micro prey from moving water by use of nets or leg hairs and specialized mouthparts; pie = piercers of filamentous algae; pre = predators; sca = scavengers of larger dead animals; shd = shredding detritivores of CPOM (mainly fallen leaves, dead plant tissue); shh = shredding herbivores of living plant tissue. ^5^ [5]. Biogeographic Regions: AT = Afrotropical, AU = Australasian, EP = East Palearctic, NA = Nearctic, NT = Neotropical, OL = Oriental, WP = West Palearctic.

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
