# Peer review of "Diversity and Ecosystem Services of Trichoptera"

_insects, 2019, doi:10.3390/insects10050125_

Round 1

Reviewer 1 Report

Morse et al. provide a well-written, beautifully illustrated description of the diversity found in aquatic Order Trichoptera, the ecosystem services that they provide, some phylogenetic context for understanding trichopteran diversity, and provide some suggestions for timing for the evolution of the ecological roles taken by some trichopteran taxa.  On the whole, I believe that this is a useful contribution, but I would like to raise several issues for the authors to consider in a revision.

Issues for consideration:

Section 3. Ecosystem Services. Suggest including reference to at least some of the important work of Scott Wissinger in this section (see [1] and references therein).

Section 5.4. Evolution of Angioseprm Phytophagy and Case Material.  This is perhaps the most controversial section of the manuscript that argues that the predicted age of the Plenitentoria Trichoptera based on molecular phylogenetic reconstruction (175 Ma, early Jurassic) is evidence for the evolution of angiosperms at that time.  This argument is based on the assumption that the modern angiosperm-based diet in most modern Plenitentoria is shared with their earliest plenitentorian ancestors.  This may or may not be a safe assumption as even modern detritivorous plentitentorian caddisflies can play other roles in ecological communities [2].  I would suggest that recently described fossils of flowers dating to 174 Ma [3] constitute better and more direct evidence that angiosperms had evolved by the early Jurrasic.

Also, relevant to the discussion (because all pairs of sister taxa are by definition the same age) of the age of the entire Order Trichoptera (estimated to be 234 Ma by Malm et al. [4]) is that the age of sister-Order Lepidoptera has been estimated by both molecular phylogenetics and fossils to be about 200 Ma [5].

Section 6. Conclusions. lines 464-470: The authors might consider including references to other molecular phylogenetic studies of Trichoptera such as the mitochondrial phylogenomics approach being taken by the Living Prairie Mitogenomics Consortium [6]

Minor corrections:  There appear to be several places where the in-text citations are in the wrong format, including lines 80-81 “(Morse, 2017)” and line 432 “(Bell et al. 2005)”.

References

1.         Wissinger, S.A.; Oertli, B.; Rosset, V. Invertebrate communities of alpine ponds. In Invertebrates in Freshwater Wetlands, Batzer, D., Boix, D., Eds. Springer: Switzerland, 2016; pp. 55-103.

2.         Wissinger, S.A.; Sparks, G.B.; Rouse, G.L.; Brown, W.S.; Steltzer, H. Intraguild predation and cannibalism among larvae of detritivorous caddisflies in subalpine wetlands. Ecology 1996, 77, 2421-2430.

3.         Fu, Q.; Diez, J.B.; Pole, M.; García Ávila, M.; Liu, Z.-J.; Chu, H.; Hou, Y.; Yin, P.; Zhang, G.-Q.; Du, K., et al. An unexpected noncarpellate epigynous flower from the Jurassic of China. eLife 2018, 7, e38827, doi:10.7554/eLife.38827.

4.         Malm, T.; Johanson, K.A.; Wahlberg, N. The evolutionary history of Trichoptera (Insecta): A case of successful adaptation to life in freshwater. Syst. Ent. 2013, 38, 459-473.

5.         Wahlberg, N.; Wheat, C.W.; Peña, C. Timing and patterns in the taxonomic diversification of Lepidoptera (butterflies and moths). PLoS ONE 2013, 8, e80875. doi: 80810.81371/journal.pone.0080875.

6.         Living Prairie Mitogenomics Consortium. The complete mitochondrial genome of the North American pale summer sedge caddisfly Limnephilus hyalinus (Insecta: Trichoptera: Limnephilidae). Mitochondrial DNA B Resour. 2019, 4, 413-415. doi: 410.1080/23802359.23802018.21547158.

Reviewer 2 Report

This is indeed an interesting review of the topic, and you have provided some informative insights into the phylogeny and ecosystem services of this group. Please see below for my suggested revisions. There are some minor grammatical errors throughout such as the use of articles (the, an/a) and pluralization. I have provided example corrections for some of these in the attached pdf version of your manuscript that should be applied throughout. In addition, there are some inconsistencies with the information from one section to another regarding the fossil evidence. I have indicated these as major revisions because this is a review of the phylogeny of this group so it is important to understand the current understanding of the fossil evidence and to be consistent when reporting it.

Minor revision:

1.      Page 1, lines 26–30: a few grammatical errors. Remember that singular nouns (e.g. infraorder, origin) are usually accompanied by an article (the or a/an). This is a persistent error throughout.

2.      Numbers under 10 should be written as words unless accompanied by a unit.

3.      Do not use both styles of in-text citations, remove (author, date) citations, e.g., Morse 2017

4.      When using the order name in front of a noun that it is describing then the adjective form “trichopteran” should be used. You would not say, e.g., “Germany taxa”, it would be “German taxa”.

5.      The figure captions for figures split into two parts (3, 6, 8, 10, 11) should be formatted as normal figure captions as per the template for this journal. As one caption, with each part described within that caption.

6.      When discussing geological ages, if you are referring to the epoch, then the name should be capitalized, e.g., “the Late Triassic”. If you are just referring generally to a later time with the period, then something like, “later Triassic” or “latest Triassic” should be used.

7.      Range should only be indicated when there is a range between two numbers, e.g., “46–47”, there is no range between these numbers.

8.      “Ansorge” not “Ansorg”.

9.      The references do not appear to be in the same font throughout. They are also not formatted consistently, please refer to the journal’s instructions for reference formatting.

Major revisions:

1.      You do not cite any literature until section 2 of the introduction. For all information that you have not gleaned from the results of your study or that is not “common knowledge” you should provide citations stating where you got that information from. There is some quite detailed information in the first section of the introduction that should have citations.

2.      Table 1 does not appear to include all of the oldest fossils known from these families, see the following comment.

3.      Lines 404­–405: Inconsistency: You state that the oldest trichopteran is Liadotaulius maior from the Toarcian, but in Table 1 you list Necrotauliidae from the Late Triassic for which Kelly et al. (2018) discussed several species older than the Jurassic. There are also several prorhyacophilids and necrotauliids from the Middle to Late Triassic of Central Asia (Nicholson et al., 2015), which are probably the oldest trichopterans and do not appear in your table. Also, Liadotaulius is provisionally considered in the family Philopotamidae (Zhang et al. 2017 [47]). You cite this paper in your study. Also, in this sentence you state that the age of L. maior from the Toarcian “compares with” that of the Malm (234 Ma), but these diverge greatly. The Toarcian was around 180 Ma, leaving 50 myr and at least one mass extinction between those estimates. Also, “Lower Toarcian” not “lower Toarcian”.

4.      When discussing the phylogeny of the order and especially the basal lineages, it would be informative to include discussion of their relationship with Lepidoptera, which they are closely related to. The basal families (e.g., Necrotauliidae) can be very difficult to distinguish from early moths, which introduces difficulties when inferring the origin of the order from fossils unless specific morphological features are found, such as a haustrum for Trichoptera (Kelly et al., 2018 [5]) or scales for Lepidoptera (van Eldijk, 2018; Zhang et al., 2018).

5.      As one of the main components of the manuscript, ecosystem services should also be returned to in the conclusion. You briefly mention some functional traits and their use within the ecosystem, but these are not clearly linked to ecosystem services (i.e. the benefit to human society). There is no major rewrite required for this, just a few brief lines linking the phylogenetic conclusions to specific ecosystem services.

References

van Eldijk, T.J., Wappler, T., Strother, P.K., van der Weijst, C.M., Rajaei, H., Visscher, H. and van de Schootbrugge, B., 2018. A Triassic-Jurassic window into the evolution of Lepidoptera. Science advances, 4(1), p.e1701568.

D. B. Nicholson, P. J. Mayhew, and A. J. Ross, “Changes to the fossil record of insects through fifteen years of discovery,” PLoS ONE, vol. 10, no. 7, Article IDe0128554, 2015.

Zhang, Q., Mey, W., Ansorge, J., Starkey, T.A., McDonald, L.T., McNamara, M.E., Jarzembowski, E.A., Wichard, W., Kelly, R., Ren, X. and Chen, J., 2018. Fossil scales illuminate the early evolution of lepidopterans and structural colors. Science advances, 4(4), p.e1700988.

Round 2

Reviewer 1 Report

I am pleased with the revisions and have no further suggestions for improvement.

Reviewer 2 Report

The authors have responded to and addressed all of my concerns from the initially submitted manuscript and I have no further comments regarding the revised manuscript, which is much improved. I'm glad that my review was of assistance and am happy to recommend that the manuscript be accepted.